

# Variations and gradients between methane seep and off-seep microbial communities in a submarine canyon system in the Northeast Pacific

Milo E Cummings[1], Lila M. Ardor Bellucci[2], Sarah Seabrook[3], Nicole A. Raineault[4], Kerry L. McPhail[5] and Andrew R. Thurber[1,2]

[1] Department of Microbiology, College of Science, Oregon State University, Corvallis, OR, United States of America
[2] College of Earth, Ocean, and Atmospheric Sciences, Oregon State University, Corvallis, OR, United States of America
[3] National Institute of Water and Atmospheric Research, Wellington, New Zealand
[4] University of South Florida, St. Petersburg, FL, United States of America
[5] College of Pharmacy, Oregon State University, Corvallis, OR, United States of America

Corresponding author
Milo E Cummings,
cumminsu@oregonstate.edu

## ABSTRACT

Methane seeps are highly abundant marine habitats that contribute sources of chemosynthetic primary production to marine ecosystems. Seeps also factor into the global budget of methane, a potent greenhouse gas. Because of these factors, methane seeps influence not only local ocean ecology, but also biogeochemical cycles on a greater scale. Methane seeps host specialized microbial communities that vary significantly based on geography, seep gross morphology, biogeochemistry, and a diversity of other ecological factors including cross-domain species interactions. In this study, we collected sediment cores from six seep and non-seep locations from Grays and Quinault Canyons (46–47°N) off Washington State, USA, as well as one non-seep site off the coast of Oregon, USA (45°N) to quantify the scale of seep influence on biodiversity within marine habitats. These samples were profiled using 16S rRNA gene sequencing. Predicted gene functions were generated using the program PICRUSt2, and the community composition and predicted functions were compared among samples. The microbial communities at seeps varied by seep morphology and habitat, whereas the microbial communities at non-seep sites varied by water depth. Microbial community composition and predicted gene function clearly transitioned from on-seep to off-seep in samples collected from transects moving away from seeps, with a clear ecotone and high diversity where methane-fueled habitats transition into the non-seep deep sea. Our work demonstrates the microbial and metabolic sphere of influence that extends outwards from methane seep habitats.

## INTRODUCTION

Oceanic methane seeps are an important component of the global methane budget, which requires careful consideration as methane is a potent greenhouse gas (*Kvenvolden, 1988*; *Reeburgh, 2007*; *Ruppel & Kessler, 2017*). Specifically, methane seeps are sites of significant biological handling of methane, as the anaerobic oxidation of methane (AOM) performed by microbes at seeps captures an estimated 80–90% of subsurface methane before it reaches the water column (*Reeburgh, 2007*; *Luff & Wallmann, 2003*). The central microbial processes at methane seeps are the paired reactions of methane oxidation and sulfate reduction, carried out primarily by a partnership between archaeal anaerobic methane oxidizers (ANME) and sulfate-reducing bacteria (SRB) (*Boetius et al., 2000*; *Orphan et al., 2002*). Seeps are also occupied by endemic fauna that form unique habitats, most commonly sulfidic polychaete worm beds, Bathymodiolin mussel beds, Vesicomyid clam beds, and Lamellibrachia tube worm bushes (*Levin et al., 2003*; *Sahling et al., 2003*; *Guillon et al., 2017*). Methane seeps and their associated microbes also provide sources of food for deep-sea creatures, ranging from seep-endemic species to mobile, commercially important species (*Levin et al., 2016*; *Seabrook, De Leo & Thurber, 2019*; *Turner et al., 2020*; *Thurber et al., 2012*). For this reason, it is important to track the "sphere of influence" of methane seeps, which can also be indicated within the microbial communities in sediment adjacent to the areas that are visibly impacted by seepage (*Levin et al., 2016*).

Seeps vary in their microbial community composition, including in their distributions of the different ANME and SRB clades. ANME-1 and ANME-2 are more abundant globally than ANME-3. ANME-2 generally predominates at cold seep sites below 20 °C (*Ruff et al., 2015*; *Wegener et al., 2016*), whereas ANME-1 is tolerant of a broader range of temperatures and can even exist at hydrothermal vents (*Vigneron et al., 2013*; *Wegener et al., 2016*). ANME-1 is also more abundant at seeps with high fluid flow and anoxic conditions, such as the carbonate chimneys of the Black Sea (*Ruff et al., 2015*; *Wegener et al., 2016*; *Treude et al., 2007*). ANME-3 appears to be more common at colder temperatures, especially at high latitudes (*e.g.*, the Arctic and Antarctic), but can also appear at cold seeps of greater depth and at mud volcanoes (*Niemann et al., 2009*; *Bhattarai et al., 2017*; *Ruff et al., 2019*). The clades of sulfate-reducing bacteria as partners of ANME also vary between different seep sites but are most commonly Desulfobacterales of the clades SEEP-SRB1 through -SRB4 (*Wegener et al., 2016*; *Petro et al., 2019*; *Waite et al., 2020*). Though ANME at seeps frequently pair with SRB partners, some clades can carry out methane oxidation without a partner, especially ANME-3 (*Vigneron et al., 2013*; *Bhattarai et al., 2017*). Seep macrofauna also have distinctive microbial communities associated with them, often as symbiotes within their tissues. The activities of macrofauna additionally affect the sediment-based microbial communities by changing physical parameters such as the penetration of oxygen and sulfide *via* bioturbation and bioirrigation (*Wallmann et al., 1997*; *Thurber et al., 2013*; *Felden et al., 2014*; *Guillon et al., 2017*).

The above variations in methane seep microbial community composition also suggest a corresponding diversity of gene functions at seeps. In addition to methane and sulfur cycling, another important area of focus is nitrogen cycling. Certain ANME are capable

of nitrogen fixation and perform high rates of nitrogen fixation at seeps, with rates dependent on methane concentration (*Dekas et al., 2014*; *Dekas et al., 2018*; *Kapili et al., 2020*). Additionally, the clade ANME-2d (also known as *Candidatus Methanoperedens* spp.) can perform its methane oxidation with nitrate as an electron acceptor rather than sulfate (*Haroon et al., 2013*; *Berger et al., 2021*). In a broader context, deep sea microbes within and beyond seeps play important roles in nitrogen cycling. The rates of nitrogen fixation, nitrification, denitrification, and anammox in the deep sea are higher than previously expected, a discrepancy which may be explained by microbial metabolism (*Pachiadaki et al., 2017*; *Wang et al., 2017*; *Suter et al., 2021*). Along with nitrogen, trace metals play an important role in deep sea microbial metabolisms. Some clades of anaerobic methane oxidizers can directly use iron and manganese as alternate electron acceptors to sulfate, notably ANME-2d (*Beal, House & Orphan, 2009*; *Leu et al., 2020*). Iron, nickel, cobalt, molybdenum, and tungsten are also integral to enzymes for central metabolism, methanotrophy, and nitrogen cycling in ANME and sulfate-reducing bacteria (*Glass & Orphan, 2012*; *Glass et al., 2014*). Since the mining of heavy metals in the deep sea is of increasing industrial interest (*Levin & Sibuet, 2012*; *Gillard et al., 2019*), it is correspondingly important to examine how microbes use these metals in their metabolisms.

The Cascadia Margin, located in the Pacific Ocean off the coast of the northwestern United States, has been an area of interest for methane seep research since the earliest discoveries of methane seeps (*Suess et al., 1985*). Methane seepage is prolific in this area, and has been occurring for thousands of years (*Joseph et al., 2013*). Recent surveys of the Cascadia Margin with single-beam and multibeam sonar have detected 3,481 methane bubble plumes, corresponding to 1,300 subsurface sources of methane fueling these seep sites (*Merle et al., 2021*). The seep habitats of the Cascadia Margin vary in geological structure, as well as macrofaunal and microbial community composition; including carbonate rock platforms, methane hydrate fields, soft sulfidic sediment, microbial mats, Vesicomyid clam beds, and Lamellibrachia tube worm beds (*Boetius & Suess, 2004*; *Seabrook et al., 2018*).

Here, six seep and non-seep sites of the Cascadia Margin were sampled to assess the biological and chemical gradients from off-seep to on-seep. For this study, we quantify the shift of microbial community and metabolic transitions between non-seep and seep sites, including based on sediment geochemistry. Two of the seep sites, Dagorlad Seep and Emyn Muil Seep, were sampled *via* comprehensive transects. These two seeps have very different gross morphology, which we interpret as an indication of the age of each seep. The stage of microbial succession is discussed in this context.

## MATERIALS & METHODS
### Study sites and sampling
Samples were collected as part of the *E/V Nautilus* expedition NA121 in September and October 2020, which focused on Grays and Quinault Canyons (46–47°N) off the coast of Washington State, USA (Table 1, Fig. 1). Sediment push cores (internal diameter 6.4 cm) were collected from various seep and non-seep sites using ROV Hercules. For two seeps,

**Table 1  Site descriptions.**

| Site | Latitude, Longitude | Depth (m) | Cores collected | Description |
|---|---|---|---|---|
| Quinault 500 m | 47.253, −125.035 | 500 | 2 | Non-seep; rocky carbonate platforms >50 m away from active seepage |
| Westmarch Seep | 47.173, −125.145 | 1310 | 2 | Seep vesicomyid clam bed with dark reduced sediment and polychaete worms |
| Quinault Canyon, ridge | 47.081, −125.058 | 1040 | 1 | Non-seep; ∼100 m deep canyon with glass sponges on ridge |
| Quinault Canyon, bottom | 47.083, −125.060 | 1120 | 2 | Non-seep; ∼100 m deep canyon (same as above); canyon base with fish and starfish present |
| Emyn Muil Seep | 46.783, −125.262 | 1070 | 5 | Seep site with thick rocky carbonate platforms; also featured areas of Calyptogena clam beds, dark reduced sediment, and small microbial mats |
| Dagorlad Seep | 47.057, −125.028 | 1010 | 6 | Seep site with extensive (∼50 m wide) microbial mat and soft sediment; vesicomyid clams on periphery of mat |
| OOI Regional Cabled Array | 44.515, −125.390 | 2900 | 2 | Slope Base site (PN1A); non-seep site near southern Hydrate Ridge |

named "Dagorlad" and "Emyn Muil," a transect-based sampling scheme was used to assess biogeochemical gradients from "off-seep" to "on-seep." This was achieved by marking the location of active methane seepage, and sampling in a line: 150 m from seep, 50 m from seep, 5 m from seep, and on-seep.

To contrast with a background sample, we also included two sediment cores from the Slope Base site of the Ocean Observatories Initiative Regional Cabled Array at 2,900 m water depth (44.5152°N, −125.3898° W). This site was located 300 km south of the other samples and was included to compare a deeper non-seep site of the same geographical region. These samples were collected by *ROV Jason* during routine maintenance of the OOI site.

Cores were extruded shipboard and sectioned into one cm-deep samples of sediment, ranging from surface (0 cm) to 10 cm deep in the core, with the exterior discarded to prevent sediment smearing from impacting the measured microbial community. These samples were frozen at −80 °C for microbiological analysis and porewater was extracted using Rhizons (0.15 μm pore size) inserted into cores at two cm intervals, often from a core collected adjacent to the one used for microbiology (if not the same core used for microbiology). Porewater was frozen at −20 °C (for major ions) or preserved in 0.05 M zinc acetate and refrigerated at 4 °C (for hydrogen sulfide). Sediment samples for methane analysis were collected in three cm sediment fractions using a sub-core, and were preserved by mixing with two mL of 5M NaOH and refrigerating upside-down at 4 °C.

## Methane, sulfide, and sulfate measurements

Ions were analyzed on an Integrion Ion Chromatograph at Oregon State University using an IonPac CS16-4 μm, 4 ×250 mm column. Methane samples were analyzed on a Picarro G2131-i Cavity Ring Down Spectrometer coupled to a Small Sample Injection Module at Oregon State University. Stabilized sulfide samples were quantified on a Shimadzu

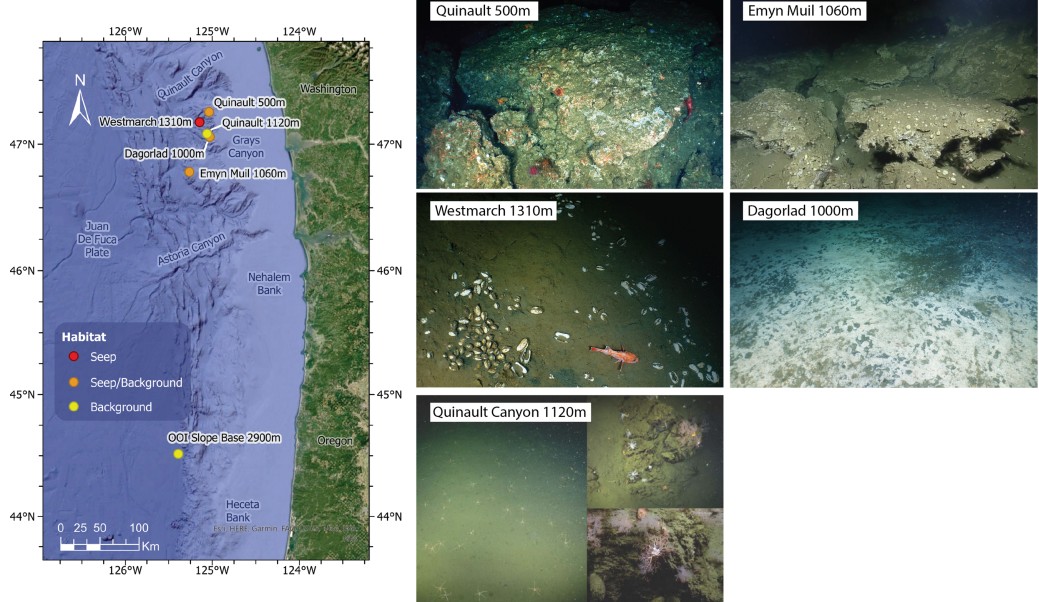

**Figure 1** **Map and seep imagery.** Map of sampling sites and ROV imagery from each site. Quinault 500 m was an active seep site at 500 m depth, but samples were only taken from 180 m and 50 m off-seep due to uncoreable rock on-seep. Westmarch Seep was a small seep at 1,310 m depth indicated by a vesicomyid clam bed. Quinault Canyon was a non-seep canyon site at 1,120 m depth (at base) with images shown from the canyon floor (left) and canyon walls (right). Emyn Muil was a seep at 1,060 m depth with extensive, deep carbonate platforms. Dagorlad Seep was a seep at 1,000 m depth with a large (>50 m wide) white and grey microbial mat. OOI Slope Base was located at the 2,900 m depth Oregon Slope Base Site of the Ocean Observatories Initiative's Regional Cabled Array; ROV imagery for this site was not available.

UV-1201S spectrophotometer at 670 nm following Cline's methylene blue method (*Cline, 1969*).

## DNA extraction and sequencing

Frozen sediment samples were extracted for DNA using the QIAgen PowerSoil kit, starting with 0.25−0.50 g of sediment and following the standard kit protocol. After DNA extraction, samples were sequenced following the Earth Microbiome protocol, wherein the V4 16S rRNA region is amplified using the improved F515/R806 primers (*Walters et al., 2015*), as detailed in a previous paper (*Seabrook et al., 2018*). Sequencing was carried out at the Oregon State University Center for Qualitative Life Sciences on the Illumina MiSeq platform, with 250-bp paired-end reads. Sequencing depth for these samples was on average 48,000 reads per sample and ranged from 1,000 to 210,000 reads.

The full set of 16S rRNA gene data was deposited in the SRA archive and is available under Bioproject PRJNA718843.

## Data analysis and statistics

Bioinformatics of the 16S rRNA analysis was carried out using a QIIME2 (v2021.2) pipeline, using DADA2 and the SILVA 16S database v138.1 (*Bolyen et al., 2019*; full pipeline provided in Data S1). We employed the PICRUSt2 program, which provides predictive gene family

profiles and functions inferred from the 16S genes present (*Douglas et al., 2020*). Statistics and figure generation, based on the outputs of the QIIME2 and PICRUSt2 analyses, were carried out in RStudio with the packages phyloseq (*McMurdie & Holmes, 2013*), vegan (*Oksanen et al., 2020*), ANCOM-BC (*Lin & Peddada, 2020*), and microbiome (*Lahti et al., 2017*). Additional figures were generated *via* SigmaPlot 14.0 (Systat Software Inc.).

For statistical analysis, the amplicon sequence variant (ASV) table output from QIIME2 was used, as well as the KEGG pathway/KO number table output from PICRUSt2. For both data sets, samples with sequencing depth below 4,000 reads were filtered out. Data were then rarefied to 4,000 and Chao1 richness metrics were calculated. Shannon alpha-diversity metrics were also calculated but found to show similar patterns to Chao1 (Table S1). Non-metric multidimensional scaling (NMDS) plots were generated using Bray-Curtis distance. The PICRUSt2 output was used to curate a database of genes (Table S2) for iron and metal transport as well as sulfur, nitrogen, and methane metabolism. This database was used to focus our analysis on particular genes and processes of interest for seep systems.

## RESULTS

### Microbial community composition

Analysis of the V4 region of the 16S rRNA gene microbial communities identified a total of 72,649 ASVs. 5,492 of these ASVs were singletons. When samples were grouped as either seep-associated (microbial mat, clam bed, ampharetid bed) or non-seep (5 m off-seep, 50 m off-seep, 150 m off-seep, or OOI non-seep sediment), there were differences between the microbial communities (Fig. 2A; PERMANOVA $P < 0.001$, pseudo-F $= 25.43$, DF $= 212$). There were 11,631 ASVs that appeared only at seep sites, including 3,539 Archaeal ASVs. At the dominant taxa level, there were also clear and gross community differences between the seep sites (Emyn Muil, Dagorlad, and Westmarch Seeps; Fig. 3) and non-seep sites. We employed an ANCOM (*via* the ANCOMBC R package) analysis to pinpoint microbial families that were statistically different between non-seep and seep sites (Fig. 4, Table S3). The bacterial families Scalinduaceae, Woeseiaceae, and NB1-j were significantly more abundant in non-seep sites *vs.* seep sites. In non-seep sites, Scalinduaceae made up on average $4.3 \pm 3.4\%$ of the microbial community, and Woeseiaceae and NB1-j made up $5.1 \pm 4.3\%$ and $4.9 \pm 2.7\%$, respectively. On the other hand, all seep sites had Sulfurovaceae, Desulfobacteraceae, and Sulfurimonadaceae as prominent families. At seep sites, Sulfurovaceae made up on average $5.0 \pm 4.9\%$ of the microbial community, and Desulfobacteraceae and Sulfurimonadaceae made up $2.8 \pm 2.6\%$ and $3.5 \pm 4.6\%$ respectively. Despite these moderate means, certain taxa were very prominent in individual samples. For example, in one microbial mat core from Dagorlad Seep, Sulfurimonadaceae made up 23.1% of the total microbial community at the surface, and Sulfurovaceae made up 24.6% of the total microbial community at three cm depth. Individual seep sites also showed variation in particular bacterial families present. Desulfosarcinaceae, the family containing SEEP-SRB1, was present at all seep sites ($8.1 \pm 5.3\%$ of the microbial community) but was particularly abundant at Emyn Muil Seep ($11.9 \pm 5.4\%$). Desulfocapsaceae, the family containing SEEP-SRB4, was present in small numbers at all seep sites ($1.8 \pm 2.4\%$), but

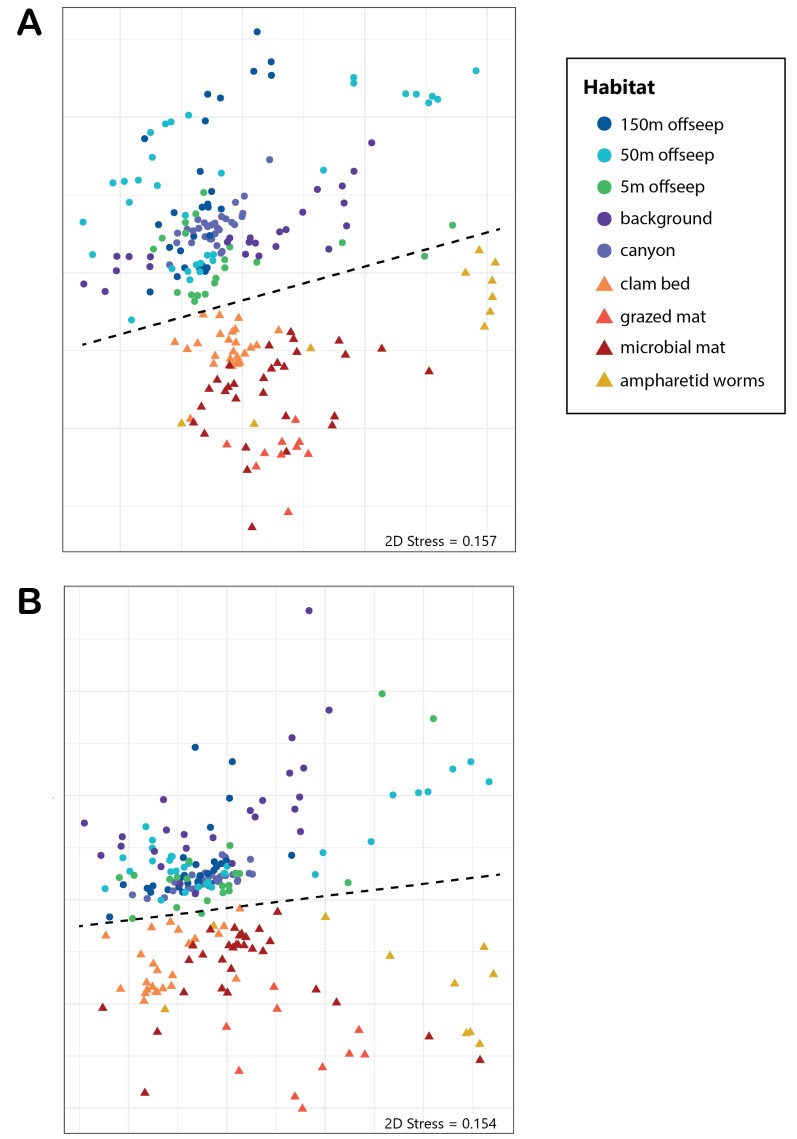

**Figure 2  NMDS of microbial diversity and PICRUSt2 functional diversity.** nMDS plots using Bray-Curtis distance of (A) overall microbial diversity, and (B) PICRUSt2-predicted functional diversity of a selected database of genes. This database consists of genes for methane, sulfur, and nitrogen metabolism, as well as metal transport and chelation genes, and is outlined in Table S1. Points that are warm-colored and triangle-shaped represent samples from seep-associated sites. Points that are cool-colored and circle-shaped represent samples from non-seep-associated sites. In both plots, seep and non-seep groupings were significantly different according to PERMANOVA, as indicated by the dashed line. ((A) $p > 0.001$, pseudo-F = 25.43, DF = 212; (B) $p > 0.001$, pseudo-F = 12.77, DF = 212).

more abundant at Westmarch ($2.2 \pm 0.8\%$). The family Desulfobulbaceae was present both at non-seep sites ($4.1 \pm 3.5\%$) and seep sites ($3.0 \pm 3.1\%$) with slightly less abundance at the latter.

Archaeal ASVs were, at maximum, 39.9% of the total microbial community at each particular site, and were more abundant at seep sites ($10.4 \pm 9.4\%$ relative abundance)

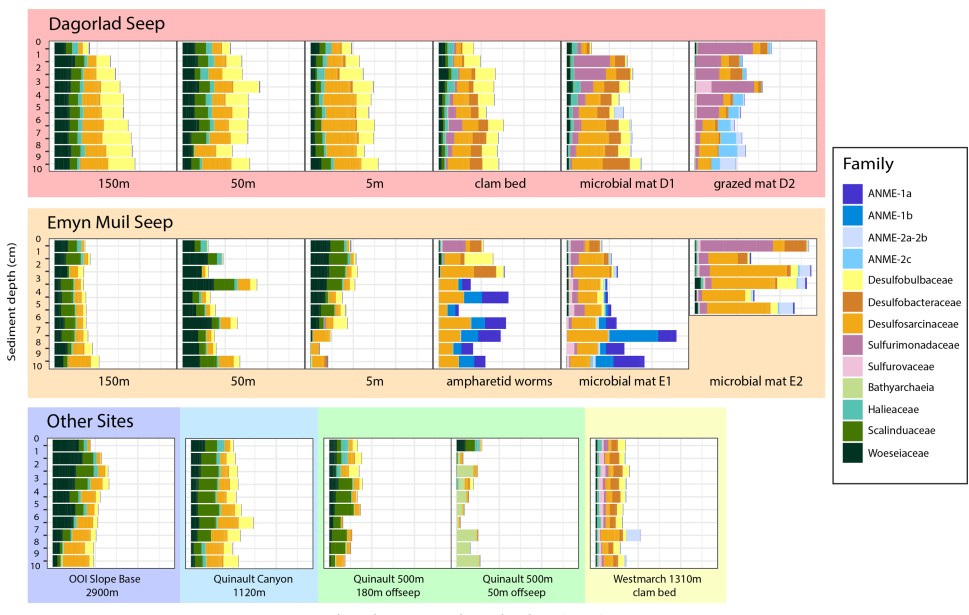

**Figure 3** Taxa barplots of microbial community relative abundance of 13 most abundant families.

compared to off seeps (6.5 ± 5.0% of the microbial community). Woesarchaeales was present at both seep and non-seep sites in about equal abundance, accounting for an average of 2.5 ± 1.5% of the microbial community. The Archaeal family Nitrosopumilaceae made up an average of 4.2 ± 3.6% of the total microbial community at the OOI sites, whereas it represented less than 1% of the microbial community at other sites. Similarly, 5m off-seep of Emyn Muil Seep, Asgard Archaea (Heimdallarchaeia and Lokiarchaeia) comprised 3.8 ± 4.1% of the total community, and were less than 1% elsewhere. Additionally, at the 500m depth Quinault sample, the prominent Archaeal members were Bathyarchaeia and Hadarchaeales, averaging 3.0 ± 3.7% and 3.1 ± 2.2% of the total microbial community, respectively.

At seeps, ANME were prominent, and the most abundant Archaeal members of the community, making up 34.3% of Archaeal ASVs at all seep sites. The clades of ANME present varied at different seeps; ANME-2 (a, b, and c) was most abundant at Dagorlad Seep, representing 15.7% of the total microbial community at nine cm depth, whereas primarily ANME-1 (a and b) was most abundant at Emyn Muil Seep, accounting for 33.1% of the total microbial community at seven cm depth. The two clam bed samples from Westmarch Seep had ANME-2a and -2b present, but these taxa comprised at most 6.8% of the total microbial community at seven cm depth, and most samples had less than 1% of the community as ANME. ANME-2d (*Candidatus Methanoperedens* spp.) and ANME-3 were not detected in any of the samples.

## Microbial and gene functional diversity

Microbial diversity, as quantified by Chao1, was highest at Quinault Canyon, averaging 1436 ± 292 (Fig. 5). The off-seep cores of Emyn Muil and Dagorlad had variable moderate

## Differential Abundance of Microbial Families, Comparing Seep Sites Against Non-seep

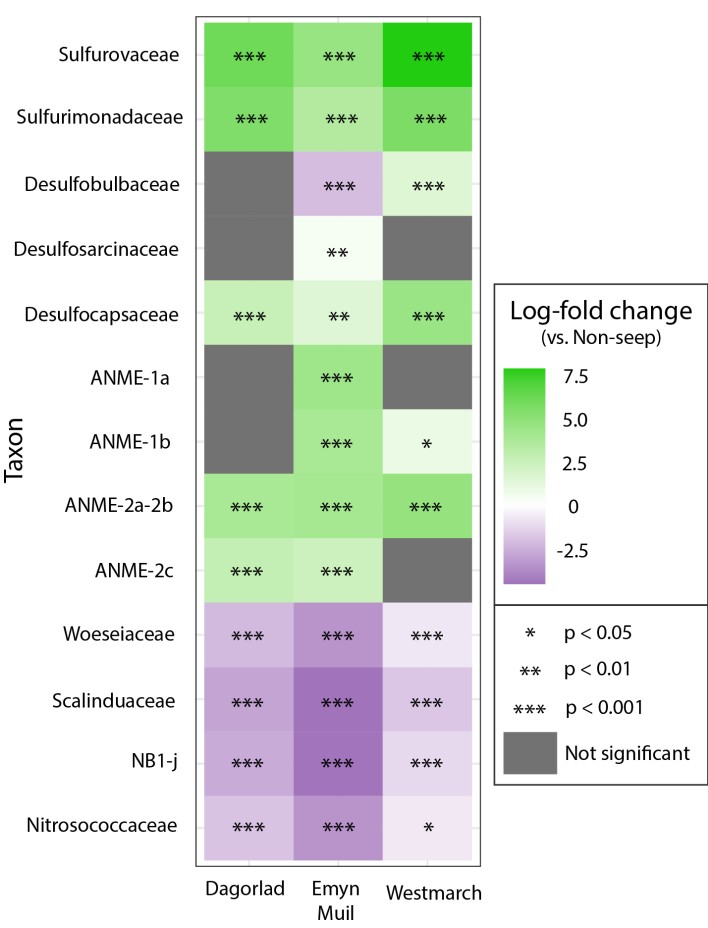

**Figure 4   ANCOM-based heatmaps of differentially abundant families between non-seep *versus* particular seep sites.** Full table of log-fold changes, *p*-values, and test statistics available in Table S2.

to high values of diversity, with a mean of 772 ± 404. On-seep cores had low diversity values, with a mean of 452 ± 204. There were local peaks in diversity at the 5m off-seep sites, with values of 861 ± 162 at this distance compared to 468 ± 199 at the adjacent distances. The OOI sample (2,900 m non-seep) had consistent low diversity values of 510 ± 53.

Predictive gene analysis resolved unique functions among the various habitats sampled. Off-seep samples were characterized by ammonia oxidation (*hao* genes) and sulfate reduction. On the other hand, on-seep samples were characterized by anaerobic methane oxidation and sulfate reduction. The sediment depths with high anaerobic methane oxidation also indicated higher abundance of nitrogen fixation genes.

Application of predictive gene function to our community data also identified clear differences between on and off seep dominant metabolism and process (Fig. 2B; PERMANOVA $p > 0.001$, pseudo-F = 12.77, DF = 212). Specifically, genes accounting

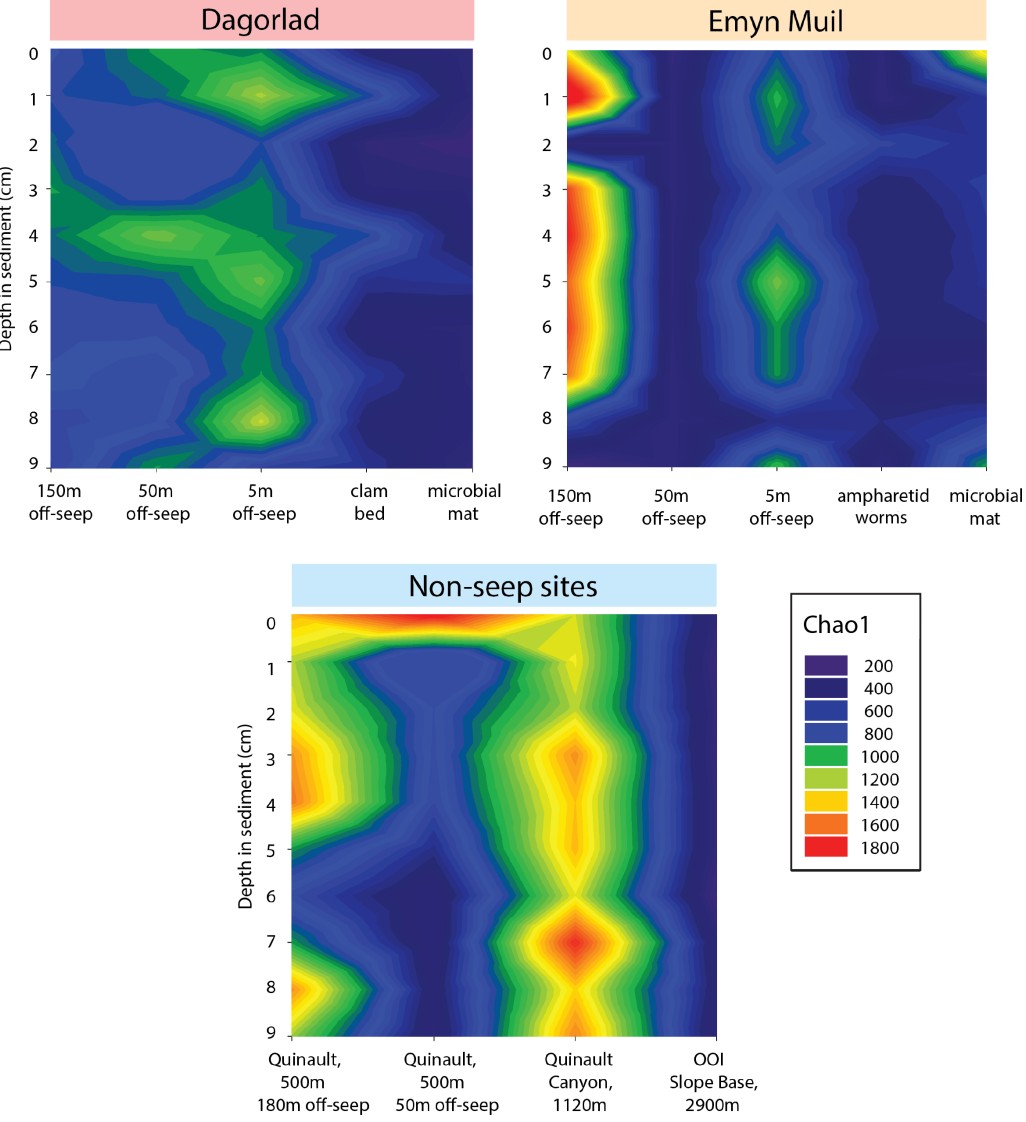

**Figure 5  Chao2 diversity of Emyn Muil, Dagorlad, and non-seep sites.** Heatmaps of Chao1 diversity at Dagorlad Seep, Emyn Muil Seep, and the non-seep sites. The plots represent microbial diversity within each sediment core, with surface-layer sediment at the top, and 10 cm into the sediment at the bottom. The seep heatmaps are arranged from off-seep to on-seep, left to right. The non-seep heatmap is arranged by increasing water depth left to right.

for up to 20% of the variance between seep and non-seep sites included genes for handling methane, sulfate reduction, nitrification, and zinc and cobalt transport (Table 2, Table S4). Methanotrophy-related genes and sulfate reduction genes were more abundant within seep samples. Additional genes (accounting for up to 40% of the variance) included other metal transport (iron, nickel, molybdenum, and tungsten).

Predicted gene function corresponded with the geochemistry within the sediment (Fig. 6). High sulfide and methane concentrations, as well as methane oxidation genes,

**Table 2  SIMPER analysis of genes contributing to up to 20% of variance between seep *vs.* nonseep sites.**

| KO | Cumulative % variance | Process | More abundant in… | Gene |
|---|---|---|---|---|
| *accounting for up to 20% of variance:* | | | | |
| K03388 | 5.87% | methanotrophy | Seep | hdrA2; heterodisulfide reductase subunit 2 |
| K03389 | 8.27% | methanotrophy | Seep | hdrB2; heterodisulfide reductase subunit 2 |
| K11261 | 10.61% | methanotrophy | Seep | fwdE,fmdE; formylmethanofuran dehydrogenase subunit E |
| K10535 | 12.89% | nitrification | Non-seep | hao; hydroxylamine dehydrogenase |
| K05601 | 15.06% | nitrification | Non-seep | hcp; hydroxylamine reductase |
| K09815 | 16.79% | zinc transport | Seep | znuA; zinc transport system substrate-binding protein |
| K03390 | 18.49% | methanotrophy | Seep | hdrC2; heterodisulfide reductase subunit C2 |
| K02190 | 20.10% | cobalt transport | Seep | cbiT; cobalt-precorrin-6B(C15)-methyltransferase |

appeared at seep-specific sites. However, these were variable between each on-seep sample from Dagorlad and Emyn Muil. Both Emyn Muil on-seep samples and one Dagorlad on-seep sample had high concentrations of sulfide throughout the core, increasing slightly with depth. The second Dagorlad on-seep (clam bed) sample had low sulfide at the surface and displayed a sharp increase in sulfide from 3 cm, with a peak below 5 cm depth. In the Dagorlad on-seep microbial mat sample, methane oxidation genes began to appear at 3 cm depth, although these genes were not as abundant overall as in the Emyn Muil cores, and methane concentrations were consistently high throughout this core. Both Emyn Muil on-seep cores had peaks in abundance of methane oxidation genes lower in the sediment, at 7 cm in the ampharetid sediment sample, and 4–5 cm in the microbial mat sample.

Sulfate and sulfate reduction genes were common at all sites. Sulfate concentrations were consistently around 1 micromolar off-seep and in the Dagorlad on-seep clam bed sample, in which sulfate reduction genes increased with sediment core depth, and methane oxidation genes were not detectable. Sulfate concentration decreased as low as 0.01 micromolar in both on-seep samples from Emyn Muil and the on-seep microbial mat sample from Dagorlad, in which a pattern of decreasing sulfate concentration below 3 cm is mirrored by an increase in sulfate reduction genes.

# DISCUSSION

## Off-seep samples and depth

Between all the core samples, the microbial community showed variation based on water depth, distance from seep, and site (especially Emyn Muil *versus* Dagorlad; Fig. 2). Non-seep sites had many community members in common, such as Desulfobulbaceae, Desulfosarcinaceae, Scalinduaceae, and Woeseiaceae, and tended to cluster together on nMDS plots. However, community composition at non-seep sites displayed some variation with depth of the sites. For instance, the shallowest site (Quinault, at 500m depth) showed smaller numbers of the sulfate-reducing bacteria Desulfobulbaceae and Desulfosarcinaceae than non-seep sites at depths 1,000 m and below. Additionally, the 50m off-seep site at Quinault showed numbers of Bathyarchaeota not present in any other core. Although it is unclear why Bathyarchaeia were present in this sample and not others, Bathyarchaeal

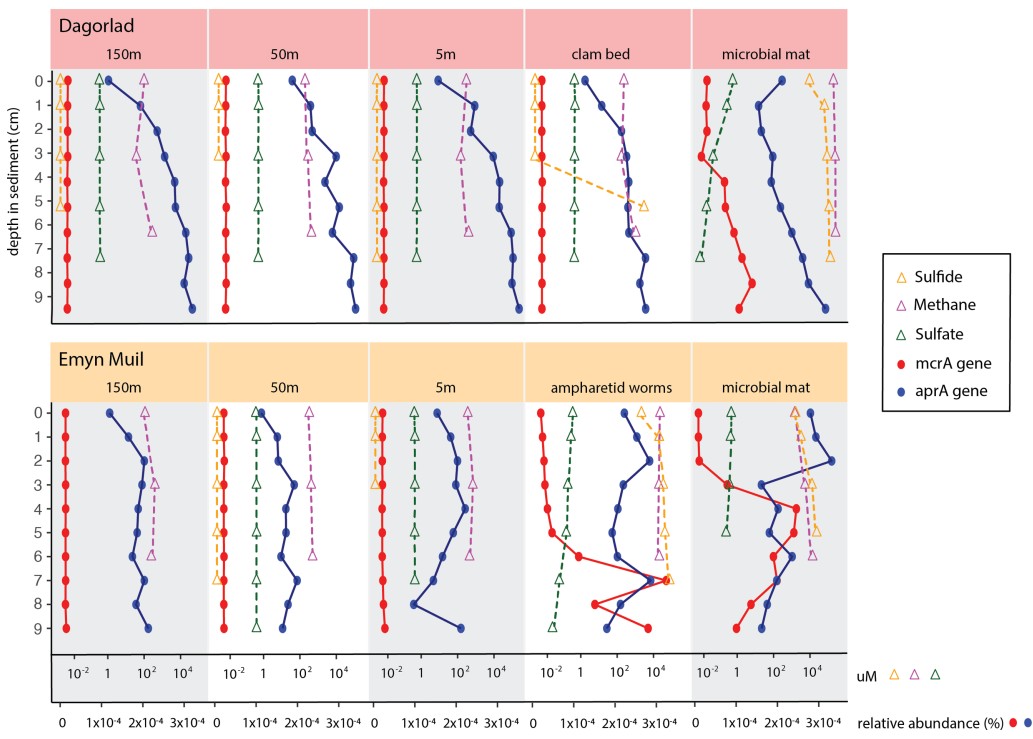

**Figure 6** Comparing geochemistry values to PICRUSt2 gene abundance at seep sites.

genomes have been shown to contain MCR (methyl coenzyme reductase) genes, indicating that they could be conducting methanogenesis or anaerobic methane oxidation (*Evans et al., 2015*). The OOI samples, which came from the deepest site at 2,900 m, had the lowest overall diversity of the samples, and also had greater numbers of Woeseiaceae and Nitrosopumilaceae. The samples from 2,900 m appeared to consist of a more specialized community whose members performed denitrification and ammonia oxidation. As such, these non-seep communities likely play important roles in nitrogen cycling within the deep ocean (*Mußmann et al., 2017*; *Wang et al., 2017*).

The detection of Archaeal ASVs in all samples is limited by the choice of sequencing primer, as all primers have biases. The original F515/R805 primers (*Caporaso et al., 2011*) did not sufficiently capture many Archaeal ASVs; however, the modified F515/R805 primers (*Walters et al., 2015*), which were used for this analysis, improved detection of *Thaumarchaeota* sequences. We intentionally chose the improved F515/R805 primers for this study because they are widely used and enable easy comparison to other studies of varying habitats. However, we recognize that our data may underrepresent some Archaeal ASVs that would be better captured by other primer sets such as ArchV34 (*Fischer et al., 2016*). Focused analysis on Archaea would benefit from the use of Archaeal-specific 16S rRNA primers or an alternate technique altogether, such as metagenome analysis. See (*Fischer et al., 2016*) for further discussion of analyzing Archaea with 16S rRNA primers.

## Differences between seep sites

Emyn Muil and Dagorlad Seeps, despite being in the same geographical region (35.9 km apart) and at similar depths (1,100 m and 1,000 m respectively) provided on-seep samples with distinct microbial communities. The sites varied significantly in terms of physical habitats. Dagorlad consisted of a soft sediment substrate covered by a large microbial mat that was 60m in diameter. On the other hand, Emyn Muil featured carbonate platforms that extended down into the sediment. Based on methane seep successional stages (*Bowden et al., 2013*), this indicates that Emyn Muil is overall a much older seep that has had time to build up these extensive carbonate platforms. Thus, one likely major factor impacting the difference between the microbial communities at these two seeps is how long the methane has been seeping at each site.

The type of ANME present at the seeps varied significantly between these two locations. At Dagorlad, ANME-2a, 2b, and 2c predominate, whereas at Emyn Muil, ANME-1a and 1b are dominant. Interestingly, in terms of ANME clades and successional stages, the opposite would be expected. Typical successional patterns of microbial communities at seeps indicate that ANME-1 is an early colonizer at seeps, which is then succeeded by ANME-2, the more globally abundant ANME clade (*Knittel & Boetius, 2009*; *Klasek et al., 2020*; *Thurber, Seabrook & Welsh, 2020*; *Knittel et al., 2005*). Therefore, a different factor likely contributes to the predominance of ANME-1 at Emyn Muil. ANME-1 is the main ANME clade present in fracture-dominated sediment with high amounts of methane fluid flow (*Briggs et al., 2011*), or in the carbonate "chimneys" of the Black Sea (*Wegener et al., 2016*). Therefore, it is likely that ANME-1 remained dominant in the Emyn Muil community due to the habitat provided by the extensive carbonate structures. Previous work from our group in the Cascadia Margin region also showed varying abundances of ANME-1 and ANME-2 at different sites (*Seabrook et al., 2018*), although in these samples, a single ANME group (1 or 2) dominated at each specific site, rather than a mix of both as we observed at Dagorlad.

Of the sulfate-reducing bacterial partners at Dagorlad and Emyn Muil, the family Desulfosarcinaceae (containing SEEP-SRB1) appeared to be the most common and was especially abundant at Emyn Muil. SEEP-SRB2 (of the Dissulfuribacteraceae family) was generally absent, or present in abundances of 1% or lower. Desulfocapsaceae/SEEP-SRB4 was not common in the samples, yet appeared at the surface of the ampharetid bed sediment sample from Emyn Muil and within the clam bed sediment sample from Dagorlad. Some research on SEEP-SRB1 has focused on its associations with ANME-2, especially regarding methods for extracellular electron transfer with the ANME partner (*McGlynn et al., 2015*; *Wegener et al., 2016*; *Skennerton et al., 2017*; *Petro et al., 2019*). Contrarily, in our samples, SEEP-SRB1 appeared at Emyn Muil where ANME-1 was most abundant. We were unable to visualize the AOM (anaerobic oxidation of methane) aggregates *in situ*, and so the specific micro-scale associations between particular ANME and SRBs elude this analysis. Nevertheless, this research implies that different combinations of ANME and SRB associates may exist within the Cascadia Margin region.

The seep site at Westmarch cannot be compared as extensively as Emyn Muil and Dagorlad because there was only a single habitat sampled (clam bed), although we can still

compare the on-seep samples of all three seep sites. While ANME were much less abundant at Westmarch Seep, in general, there was a distinct community below seven cm depth that included ANME-2a and -2b. The appearance of ANME-2a and -2b deeper in the core could suggest bioturbation by the clams, which would oxidize the sediment deeper, depressing the sulfate methane transition zone and the depth of methane oxidation. Westmarch Seep also had consistent Sulfurovaceae throughout the cores, in contrast to Dagorlad or Emyn Muil, where Sulfurovaceae was either not present or present at one or two core depths. Interestingly, though Desulfocapsaceae/SEEP-SRB4 was not common across all samples, it appeared in the Westmarch Seep samples. As discussed above, the cores from Dagorlad and Emyn Muil that contained SEEP-SRB4 were faunal habitat sediment samples. In general, when SEEP-SRB4 appears at methane seeps, it is uncommon but most often associated with ANME-2 or -3 rather than ANME-1 (*Ruff et al., 2015*; *Li, Yang & Zhou, 2020*). However, SEEP-SRB4 can appear without ANME partners at areas with larger hydrocarbons present such as mud volcanoes, and it is suggested that this group plays a role in the degradation of larger hydrocarbons (*Kleindienst et al., 2012*; *Petro et al., 2019*). At Westmarch Seep, SEEP-SRB4 and ANME-2 indeed co-occur, but this trend was not consistent at the other seeps. However, across all of our seep samples, the more common pattern is the appearance of SEEP-SRB4 in seep-associated faunal sediments; perhaps there is an unexplored link here between SEEP-SRB4, seep-associated fauna, and type of hydrocarbon fueling the seep. While we did not expect to find any longer chain hydrocarbons beyond methane, we also did not specifically quantify it.

## "Transitional" samples and the methane seep sphere of influence

The "marginal" seep samples of ampharetid worm beds and vesicomyid clam beds, as well as the 5m off-seep samples, show some interesting patterns in microbial communities and diversity that indicate a transition between non-seep and seep habitats. The ampharetid worm bed core from Emyn Muil overall grouped with other seep-associated samples, and also formed a smaller grouping, as seen to the bottom-right side of the nMDS plots (yellow triangle symbols, Fig. 2). ANME-1 was clearly present deeper in this sample, but there were few Sulfurimonadaceae present in core depths below one cm, unlike all other seep sites. The ampharetid bed samples thus appeared to have a microbial community capable of seep-associated anaerobic oxidation of methane, but different from other seep-associated samples. Samples from the Dagorlad clam bed core (orange triangle symbols, Fig. 2), clustered with other seep-associated samples but were closer in community composition to non-seep samples than to other seep samples. This pattern appears to be mirrored by the 5m off-seep samples within the nMDS plots (green circle symbols, Fig. 2). The 5 m samples clustered overall with non-seep samples but were generally closer in composition to the seep-associated samples. Of note, these patterns apply both in terms of microbial community composition, as well as targeted metabolic similarity of the PICRUSt2 database. That is to say, the 5m off-seep samples and the clam bed samples also had similar metabolic profiles in terms of sulfur, nitrogen, and methane metabolism, and metal transport. With regard to diversity, the 5m off-seep samples also showed local maxima in diversity values, positioning them between seep and non-seep in terms of community members and

metabolisms. This observation illustrates that there is a gradient of community composition and metabolism moving from on-seep to off-seep that increases local microbial diversity.

The patterns in microbial diversity tended to follow expected patterns of faunal diversity. Previous patterns in macrofaunal diversity also exhibit similar trends moving from on-to off-seep, where the zones of transition between seep and non-seep contained both seep-endemic as well as non-seep-specialized organisms, increasing overall species diversity (*Levin et al., 2010*). Intermediate habitats that transition between different specialized habitats tend to have higher diversity of fauna. For example, submarine canyons host highly heterogeneous marine habitats, and frequently have a large amount of faunal diversity (*Ramirez-Llodra et al., 2010*; *De Leo & Puig, 2018*). It is important to note that all of our samples from this study came from within canyons, and that, these samples showed some of the greatest diversity values amongst all seep samples we have collected (*Seabrook et al., 2018*; *Thurber, Seabrook & Welsh, 2020*). Within these canyon-specific samples, microbial diversity and faunal diversity appear to be related, indicating that the unique ecosystems of methane seeps impact surrounding areas in terms of both microbial and faunal diversity (*Levin et al., 2016*).

## Biogeochemistry and metabolic analysis

Application of combined gene prediction and biogeochemistry elucidated some clear patterns within these briefly visited seeps. Predicted gene abundances showed that sulfate reducing and sulfide oxidizing genes predominated at shallower core depths, and methane oxidizing genes predominated at deeper core depths. Unsurprisingly, predicted anaerobic methane oxidizing genes increased where the sediment sulfate concentrations began to drop, indicative of a sulfate-methane transition zone (*Ruff et al., 2015*).

Seep samples had higher relative abundance predictions of zinc, cobalt, nickel, and tungsten-handling proteins, and these contributed to major differences in the predicted gene composition between seep and non-seep samples (Table 2, Table S4). Nickel and zinc are needed for key proteins involved in methanotrophy (*Glass & Orphan, 2012*), and so acquisition of these metals is particularly important to ANME at seeps. Cobalt and tungsten are necessary for enzymes involved in core metabolism, but often precipitate in sulfidic seep sediment and are less bioavailable (*Glass et al., 2014*). Therefore, on-seep microbes logically need to prioritize acquisition of what little cobalt and tungsten are available. Thus, our data reflect biogeochemical trends in metal availability particular to methane seep systems.

Application of predictive gene pipelines was surprisingly informative for methane seeps. Although gene abundances were not directly measured in our analysis, and instead estimated based on 16S rRNA gene data using the PICRUSt2 pipeline, the correspondence between the measured geochemical values and the PICRUSt2 genes indicates that this analytical approach is useful even in deep-sea chemosynthetic ecosystems. Additionally, the PICRUSt2 analysis can reflect genuine trends in metabolism by summarizing genes of particular interest across whole communities. We found PICRUSt2 analysis useful for summarizing the metabolic potential of different microbial communities and revealing the transition of seep-associated metabolisms to non-seep-associated metabolisms.

## CONCLUSIONS

This work focused on the microbial communities of methane seeps from geographically close sites in the Northeastern Pacific Ocean. The microbial communities distant from methane seeps were distinct in both diversity and metabolic capability from on-seep microbial communities. The communities between methane seeps and off-seep sites indicated that the microbial influence of seeps extends outwards and creates transitional communities, or ecotones of community overlap. Off-seep communities differed from each other based on depth, whereas on-seep communities differed from each other based on gross seep morphology, which indicated the seep succession state and age were likely deterministic of the microbial community composition. Finally, applying predictive gene pipelines to these samples was highly informative, indicating transitions in metabolic capability from off-seep to on-seep. Overall, methane seeps contribute to both local-scale ocean habitats and the global carbon budget, and this work demonstrates how the microbial communities at seeps factor into these broader systems with a sphere of influence on biodiversity that extends beyond the seep habitats themselves.

### Funding

This research was funded by NOAA Ocean Exploration Grant NA19OAR0110301 and NSF Grants 2046800 and 1933165. The funders had no role in study design, data collection and analysis, decision to publish, or preparation of the manuscript.

### Grant Disclosures

The following grant information was disclosed by the authors:
NOAA Ocean Exploration Grant: NA19OAR0110301.
NSF Grants: 2046800, 1933165.

### Competing Interests

The authors declare there are no competing interests.

### Author Contributions

- Milo E Cummings conceived and designed the experiments, performed the experiments, analyzed the data, prepared figures and/or tables, authored or reviewed drafts of the article, and approved the final draft.
- Lila M. Ardor Bellucci performed the experiments, analyzed the data, prepared figures and/or tables, authored or reviewed drafts of the article, and approved the final draft.
- Sarah Seabrook conceived and designed the experiments, analyzed the data, authored or reviewed drafts of the article, and approved the final draft.
- Nicole A. Raineault performed the experiments, authored or reviewed drafts of the article, and approved the final draft.
- Kerry L. McPhail analyzed the data, authored or reviewed drafts of the article, and approved the final draft.

- Andrew R. Thurber conceived and designed the experiments, performed the experiments, analyzed the data, authored or reviewed drafts of the article, and approved the final draft.

## Field Study Permissions

The following information was supplied relating to field study approvals (i.e., approving body and any reference numbers):

Ocean Exploration Trust: E/V Nautilus expedition NA121.

## DNA Deposition

The following information was supplied regarding the deposition of DNA sequences:

The full 16S rRNA gene data is available in the NCBI Sequence Read Archive: PRJNA718843.

## Data Availability

The full pipelines for QIIME2 and PICRUSt2 analyses are available in the Supplementary File.

## Supplemental Information

Supplemental information for this article can be found online at http://dx.doi.org/10.7717/peerj.15119#supplemental-information.

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
