# Peer review of "Variations and gradients between methane seep and off-seep microbial communities in a submarine canyon system in the Northeast Pacific"

_PeerJ, doi:10.7717/peerj.15119_

## Round 0.1 · original submission · Minor Revisions

Dear authors

Congratulations on this excellent work which will make a nice contribution to the field of Deep-Sea Biology. The reviewers also enjoyed your work and made minor suggestions that you may find useful. I am looking forward to seeing your revised work.

Best regards
Angelo Bernardino

·

Basic reporting

This manuscript describing microbial communities and associated inferred metabolisms does well in describing the transition between seep and non-seep habitats. The references were appropriate and the writing was very good. Figures and Tables are well done and the data has been posted to a publically available database. Described patterns fit well with the perceived conclusions.

Experimental design

This was also well done with multiple samples coming from two of the seep habitats. Data analysis pipeline was thorough. My only question was to how well these data represent archaeal taxa as the primers used in SSU amplicon sequencing are primarily targeting bacterial taxa instead. You might add a note that though the archea were present they most likely do not represent a complete analysis of all the archaea taxa present.

Validity of the findings

Conclusion were well stated and did not over extend beyond the data as presented.

Additional comments

Nicely done both in terms of experimental design and technique application.

Reviewer 2 ·

Basic reporting

In the manuscript “Variations and gradients between methane seep and off-seep microbial communities in a submarine canyon system in the Northeast Pacific” Cummings and co-authors study the microbial communities in a submarine system in the Northeast Pacific to quantify the scale of seep influence on biodiversity within this habitat
Overall, I consider this manuscript does what is proposed in the objectives and in an appropriate way. Nevertheless, I have some minor comments that I think will improve the manuscript.

Experimental design

Material and Methods: how far (Km) was the sediment collected at the extra site? It is not clear each portion of the sediment the authors used for DNA extraction… the corers were sectioned into 1 cm, but then what was used for DNA extraction?
Data analysis: the authors calculated the CHAO (estimator for alpha diversity) but I think more alpha diversity should be calculated (H’; J’) and also beta-diversity to asses better the differences between sites

Validity of the findings

Results: How many singletons were observed? I think that is not mentioned: Also overall how many were ASV were exclusive from seep sites (general and Archaea)?
If the authors analysed different depths (my thoughts after reading the discussion) how were the taxa barplots generated for each site (averages?)

---

## Round 0.2 · accepted · Accept

Dear authors

Thank you for addressing all queries from the reviewers and I am pleased to accept this excellent work for publication.

Best regards